# Insights into the zebrafish left–right organizer's centrosomes and cilia via volume electron microscopy

Favour Ononiwu[1,2,*], Albert Lawrence Adhya[2,3], Melissa Mikolaj[4,5], Christopher Dell[4,5], Abdalla Wael Shamil[1], Kedar Narayan[4,5] and Heidi Hehnly[1,2,‡]

## ABSTRACT

The zebrafish left–right organizer (LRO), Kupffer's vesicle (KV), is a ciliated epithelial organ for which its three-dimensional architecture underlies symmetry breaking during embryonic development. While KV cilia have been extensively studied by light microscopy, their ultrastructural organization, heterogeneity, and spatial patterning within the intact organ remain incompletely defined. Here, we establish volumetric electron microscopy (vEM) as a platform for high-resolution, three-dimensional analysis of KV architecture. Using vEM, we reconstructed nearly an entire KV at nanometer resolution, enabling comprehensive assessment of cilia, centrioles, appendages, rootlet fibers, and associated vesicles within their native tissue context. Ultrastructural analysis revealed heterogeneity in centrosome architecture associated with KV cilia, including variability in centriole composition and the presence of distal and subdistal appendages as well as rootlet fibers. In addition, cilia were frequently associated with distinct classes of membrane-bound vesicles, including ciliary-associated vesicles and ciliary-associated dense vesicles. Beyond describing KV ultrastructure, this dataset illustrates how vEM can be leveraged, while also highlighting important caveats related to sampling depth, developmental staging, and interpretation of centriole loss versus remodeling. Collectively, this work provides a foundational vEM resource for the zebrafish LRO and establishes a framework for integrating volumetric ultrastructural analysis with developmental and functional studies of ciliated tissues.

KEY WORDS: Left–right organizer, Cilia, Centrosome, Volume electron microscopy, Zebrafish

## INTRODUCTION

The establishment of left–right (LR) asymmetry is a fundamental feature of vertebrate development and is essential for correct organ positioning and morphogenesis. In vertebrates, symmetry breaking is initiated within a transient, ciliated organ known as the left–right organizer (LRO). In zebrafish, this structure is the Kupffer's vesicle (KV), a fluid-filled epithelial organ functionally analogous to the mammalian node. Motile cilia projecting into the KV lumen generate a directional fluid flow that provides spatial information required for asymmetric gene expression and downstream LR patterning (Dasgupta and Amack, 2016; Grimes and Burdine, 2017).

Although the requirement for cilia-driven flow in LR axis specification is well established (Nonaka et al., 1998), the three-dimensional organization and ultrastructural diversity of cilia within the LRO – resolved at the level of electron microscopy – remain poorly defined. Much of our current understanding of KV architecture derives from immunofluorescence and live imaging approaches, which have been indispensable for mapping molecular localization, ciliary motility, and tissue-scale dynamics. However, these methods primarily report on protein presence and relative positioning, rather than on ultrastructural assembly. As a result, features such as centriole composition, appendage architecture, rootlet organization, and ciliary vesicles cannot be directly resolved, and molecular markers are often used as proxies for structural identity.

This distinction is particularly relevant during early LRO morphogenesis, when progenitor cells undergo coordinated epithelialization to form a cyst-like structure in zebrafish (Essner et al., 2005) or a cup-shaped epithelium-like structure in mammals (Lee and Anderson, 2008). While theoretical and experimental studies have provided deep insight into how ciliary motility generates directional flow (Layton, 1976; McGrath et al., 2003; Okada et al., 1999), comparatively little is known about how cilia are structurally integrated into the developing LRO epithelium. It remains unclear whether LRO cilia are ultrastructurally uniform or whether distinct subtypes exist, defined by differences in centriole number, appendage composition, or cytoskeletal anchoring.

In ciliated cells, the centrosome, comprising a mother and daughter centriole, resides at the base of the cilium, where the mother centriole serves as the basal body that templates axoneme formation (Vertii et al., 2016a). This function depends on specialized structural features, including distal appendages (DAs), which mediate docking of the basal body to the plasma membrane and license ciliogenesis (Burke et al., 2014; Joo et al., 2013; Tanos et al., 2013; Ye et al., 2014), and subdistal appendages (SDAs), which anchor cytoplasmic microtubules and contribute to basal body positioning and orientation (Ishikawa et al., 2005; Kunimoto et al., 2012; Tateishi et al., 2013). Despite the central role of centrosomes in cilia formation, it remains unclear if centrosome architecture is remodeled during LRO maturation, whether centriole elimination or structural transformation occurs in this tissue, and how variability in appendage composition might contribute to ciliary heterogeneity. Resolving these questions requires ultrastructural approaches capable of simultaneously visualizing centrosome ultrastructure within their native three-dimensional tissue context.

Volumetric electron microscopy (vEM) offers a unique opportunity to address these gaps by enabling three-dimensional ultrastructural

[1]Department of Biology, Syracuse University, Syracuse, NY 13210, USA. [2]BioInspired Institute, Syracuse University, Syracuse, NY 13210, USA. [3]Chemistry Department, Syracuse University, Syracuse, NY 13210, USA. [4]Center for Molecular Microscopy, Center for Cancer Research, National Cancer Institute, National Institutes of Health, Bethesda, MD, USA. [5]Cancer Research Technology Program, Frederick National Laboratory for Cancer Research, Frederick, MD, USA. *Present address: Division of Biological Sciences, University of California, San Diego, La Jolla, CA 92093, USA.

‡Author for correspondence (hhehnly@syr.edu)

K.N., 0000-0001-7982-6494; H.H., 0000-0001-6660-5254

Biology Open

reconstruction of entire organs at nanometer resolution (Castranova et al., 2025; Cheng et al., 2019; Insinna et al., 2019; Micheva and Smith, 2007). Beyond descriptive analysis, vEM enables quantitative interrogation of tissue architecture, including measurements of cilia number and length, centriole and appendage prevalence, rootlet frequency, spatial distribution of structural subtypes, and associations between cilia and cilia-associated vesicles. When combined with full-volume reconstructions, these measurements allow structural features to be analyzed in relation to cell position, epithelial topology, and lumen geometry. These are relationships that are difficult or impossible to infer from two-dimensional electron microscopy sections or light microscopy alone.

At the same time, interpretation of vEM datasets requires careful consideration of inherent limitations. Ultrastructural snapshots represent fixed developmental time points and cannot directly capture dynamic processes such as intraflagellar transport, centriole remodeling, or membrane trafficking. Apparent absence of centrioles or appendages may reflect true elimination, transient remodeling, or limitations in sampling depth and segmentation. Additionally, developmental heterogeneity across embryos and within individual KVs necessitates cautious extrapolation from single-volume reconstructions. Recognizing these caveats is essential for integrating vEM-based observations with functional and live-imaging approaches moving forward.

Here, we apply vEM to generate a comprehensive three-dimensional ultrastructural map of the zebrafish KV. By explicitly quantifying ciliary and centrosomal features within their native tissue context, and by outlining both the analytical power and interpretive constraints of a volumetric ultrastructural dataset, we establish vEM as a platform for studying LRO organization.

## RESULTS

### Variable Rootletin recruitment to KV basal bodies motivates ultrastructural analysis of ciliary rootlets

To determine whether cilia within the KV are molecularly uniform or exhibit heterogeneity at their base, we first assessed the localization of core ciliary and centrosomal components using immunofluorescence microscopy. As a baseline, we examined γ-tubulin as a marker of the centrosome and IFT88 as an essential component of the intraflagellar transport machinery (Bettencourt-Dias et al., 2011; Dasgupta and Amack, 2016; Pazour et al., 2000; Vertii et al., 2016b), both of which are expected to be present at functional cilia. In parallel, we analyzed the localization of Rootletin (Yang et al., 2002), the primary structural component of ciliary rootlets (van Hoorn and Carter, 2024), to test whether rootlet-associated proteins are uniformly recruited across KV cilia.

Ciliary rootlets are striated cytoskeletal filaments that extend from the basal body into the cytoplasm and are composed largely of the coiled-coil protein Rootletin (Mahen, 2021; van Hoorn and Carter, 2024). In primary cilia, rootlets contribute to basal body anchoring and mechanical stability (Antoniades et al., 2014; Lechtreck and Melkonian, 1998; Soh et al., 2020), while in motile cilia they have been implicated in basal body positioning, coordination, and resistance to mechanical stress during beating (Basquin et al., 2019; Galati et al., 2014; Jerka-Dziadosz et al., 1995). Despite these proposed roles, whether ciliary rootlets are a uniform or variable feature of vertebrate left–right organizers has not been established.

To examine the molecular organization of KV cilia, embryos at 12 h post fertilization (hpf) were immunostained for acetylated tubulin (Ac-tub) to label ciliary axonemes, γ-tubulin to mark centrosomes, IFT88 to label intraflagellar transport machinery, and Rootletin to mark rootlet-associated protein (Fig. 1A). Confocal

imaging revealed robust Ac-tub labeling of elongated cilia projecting into the KV lumen, with γ-tubulin localized to discrete puncta at the ciliary base (Fig. 1B). IFT88 consistently localized to the ciliary base and along the axoneme in all KV cilia examined (Fig. 1C). In contrast, Rootletin exhibited more variable localization: while it was frequently detected adjacent to γ-tubulin-positive centrosomes, consistent with rootlet-associated positioning, it was also observed as diffuse or punctate signal throughout the cytoplasm of KV cells (Fig. 1D).

To quantify this variability, we assessed the frequency of component association across three independent embryo clutches. All Ac-tub-positive cilia were associated with γ-tubulin and IFT88, indicating a conserved centrosomal and transport framework. However, only a subset of γ-tubulin-positive centrosomes exhibited detectable Rootletin signal, with a mean association frequency of 60.33±23.78% (Fig. 1E). This variability across clutches suggests heterogeneity in Rootletin recruitment to KV basal bodies.

Importantly, immunofluorescence detection of Rootletin does not directly report on the presence, length, or organization of rootlet filaments themselves. Rootletin localization may reflect partial assembly, transient recruitment, or non-rootlet pools of the protein. Thus, while these data reveal molecular heterogeneity at the ciliary base, they cannot resolve whether KV cilia differ in the presence or architecture of rootlet structures.

This ambiguity provided a key motivation for subsequent vEM analysis. By directly visualizing basal bodies and associated filamentous structures within the intact KV epithelium, vEM enables definitive assessment of whether ciliary rootlets are present and allows molecular variability observed by immunofluorescence to be evaluated in an ultrastructural context.

### Workflow overview for vEM imaging of the KV

To describe the potential ultrastructural heterogeneity of KV cilia, we employed vEM. vEM is a high-resolution imaging technique that combines serial ultrathin sectioning with scanning electron microscopy (SEM) followed by computational reconstruction to generate three-dimensional nanoscale maps of cellular architecture. Unlike conventional confocal or super-resolution microscopy, which is limited by optical diffraction and antibody penetration, vEM enables the visualization of fine subcellular features such as basal bodies, rootlets, and ciliary membrane specializations with nanometer precision. This technique has been successfully applied to study complex tissue architectures and organelle organization in diverse systems, including cilia and centrosomes (Castranova et al., 2025; Cheng et al., 2019; Insinna et al., 2019; Micheva and Smith, 2007). By applying this approach to the zebrafish KV, we aimed to resolve previously uncharacterized features of centrosome and cilia architecture.

To characterize KV cilia at ultrastructural resolution, we employed a specific workflow. Sox17:GFP-CAAX transgenic zebrafish embryos were screened at ∼12 hpf, and ones at the six-somite stage with identifiable GFP-positive KVs with fully formed lumens that contained between 50 and 70 KV cells were selected for processing (example shown in Fig. 2A). This selection criterion was used because KVs with substantially fewer or greater numbers of cells have been associated with LR asymmetry defects, whereas this range is most frequently observed in phenotypically normal embryos within a healthy clutch (Gokey et al., 2016). Embryos meeting these criteria were fixed, and the KV was microdissected from the rest of the embryo for downstream processing. Dissected KVs were mounted on gridded coverslips to enable spatial registration and tracking throughout the image reconstruction pipeline (Fig. 2B).

Biology Open

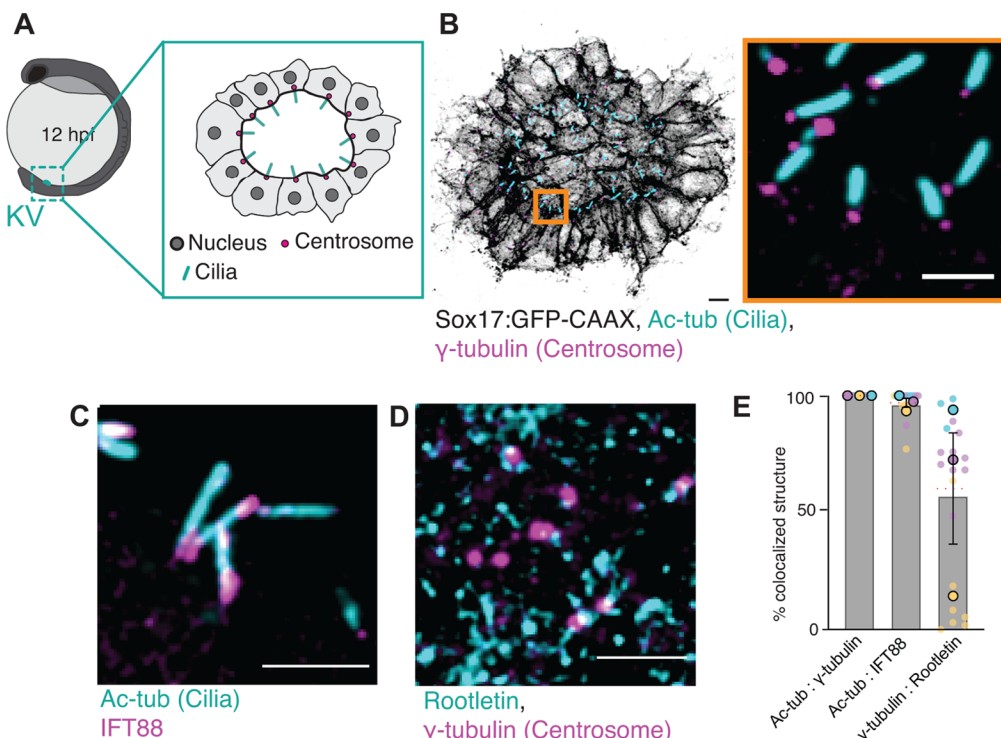

**Fig. 1. Variable Rootletin recruitment to KV basal bodies motivates ultrastructural analysis of ciliary rootlets.** (A) Schematic of a whole zebrafish embryo with an inset showing KV cells at 12 hpf. The inset illustrates the KV lumen surrounded by epithelial cells with nuclei (gray), centrosomes (red), and cilia (blue). (B) Confocal image of KV cells from *Sox17:GFP-CAAX* embryos (membranes, inverted grayscale) immunostained for acetylated tubulin (Ac-tub; cyan), marking cilia, and γ-tubulin (magenta), marking centrosomes. Inset shows a magnified view of cilia and centrosomes. Scale bars: 10 μm (main), 5 μm (inset). (C,D) Higher-magnification images showing colocalization of acetylated tubulin (cyan) with IFT88 (magenta) along the cilium (C) or γ-tubulin (magenta) with Rootletin (cyan) at the centrosome (D). Scale bars: 5 μm. (E) Quantification of colocalization events: percentage of cilia (Ac-tub+) that also contain γ-tubulin (Ac-tub:γ-tubulin) or IFT88 (Ac-tub:IFT88), and percentage of γ-tubulin-positive centrosomes colocalized with Rootletin (γ-tubulin:Rootletin). Data represent mean±s.e.m. of individual embryos, with data points color coded by clutch (*n*=3 clutches). Sample sizes: Ac-tub:γ-tubulin (*n*=29 embryos), Ac-tub:IFT88 (*n*=18), γ-tubulin:Rootletin (*n*=21). Refer to Table S1.

The samples were embedded in resin with the KV positioned near the block surface to facilitate sectioning. Block face imaging relative to the grid confirmed correct positioning (Fig. 2C). To ensure accurate internal orientation and determine the depth of KV positioning within the embryo for subsequent sectioning, we performed X-ray microscopy. This technique is increasingly used to study mouse embryonic development, as it enables acquisition of high-contrast, high-resolution datasets of whole embryos (Handschuh and Glösmann, 2022). In our study, X-ray microscopy was used to segment major structures, including the embryonic cells (green in Fig. 2D), yolk cells (orange in Fig. 2D), and KV (blue arrow in Fig. 2D). These images confirmed the internal morphology of the KV and facilitated more accurate embryo alignment for optimal sectioning (Fig. 2D,E). Ultrathin sectioning was performed using a diamond knife to generate ribbons of ~70 nm sections across the volume of the KV (Fig. 2E). These serial sections were imaged and computationally aligned to produce a volumetric reconstruction of the entire KV. Representative cross-sectional views from slices 21, 222, and 388 illustrate the well-preserved morphology and continuity of the KV structure (Fig. 2F; Movie 1). Owing to the time-intensive and technically demanding nature of this pipeline, from embryo preparation and X-ray microscopy to serial sectioning, imaging, and computational alignment, we successfully generated one complete volumetric dataset of an entire KV. All subsequent analyses are based on this single, high-quality dataset, highlighting both the strength and the current limitation of this approach.

## vEM of the KV reveals that most cilia associate with both mother and daughter centrioles, but a subset lack at least one

To establish how vEM can be used to interrogate centrosome–cilia organization within the intact zebrafish LRO, we performed comprehensive three-dimensional segmentation of a mature KV from the vEM dataset described in Fig. 2. This analysis was designed to assess what structural features can be reliably quantified from a single high-resolution volume, and to define both the interpretive power and the limitations of this approach.

Image stacks were imported into Dragonfly (Object Research Systems), a software platform optimized for visualization, segmentation, and quantitative analysis of large volumetric microscopy datasets (D'Imprima et al., 2023). Using manual and semi-automated segmentation workflows, we reconstructed the full KV volume and annotated nuclei, cilia, mother and daughter centrioles, the epithelial boundary, and the central lumen (Fig. 3A; Movie 2). All segmentations were performed across the complete series of serial 70 nm sections spanning the full KV lumen and most of the KV volume, enabling three-dimensional reconstruction and analysis. Segmentation followed a stepwise workflow in which KV cell nuclei were first segmented to identify ciliated cells, followed by segmentation of all cilia, and finally assessment of mother and daughter centrioles within those cells through segmentation of all identifiable centrioles. This volumetric reconstruction defined the overall dimensions of the mature KV (83 μm×79.5 μm×40.7 μm along the x-, y-, and z-axes, respectively)

**A** **Stereo Image**  **D** **X Ray Microscopy**

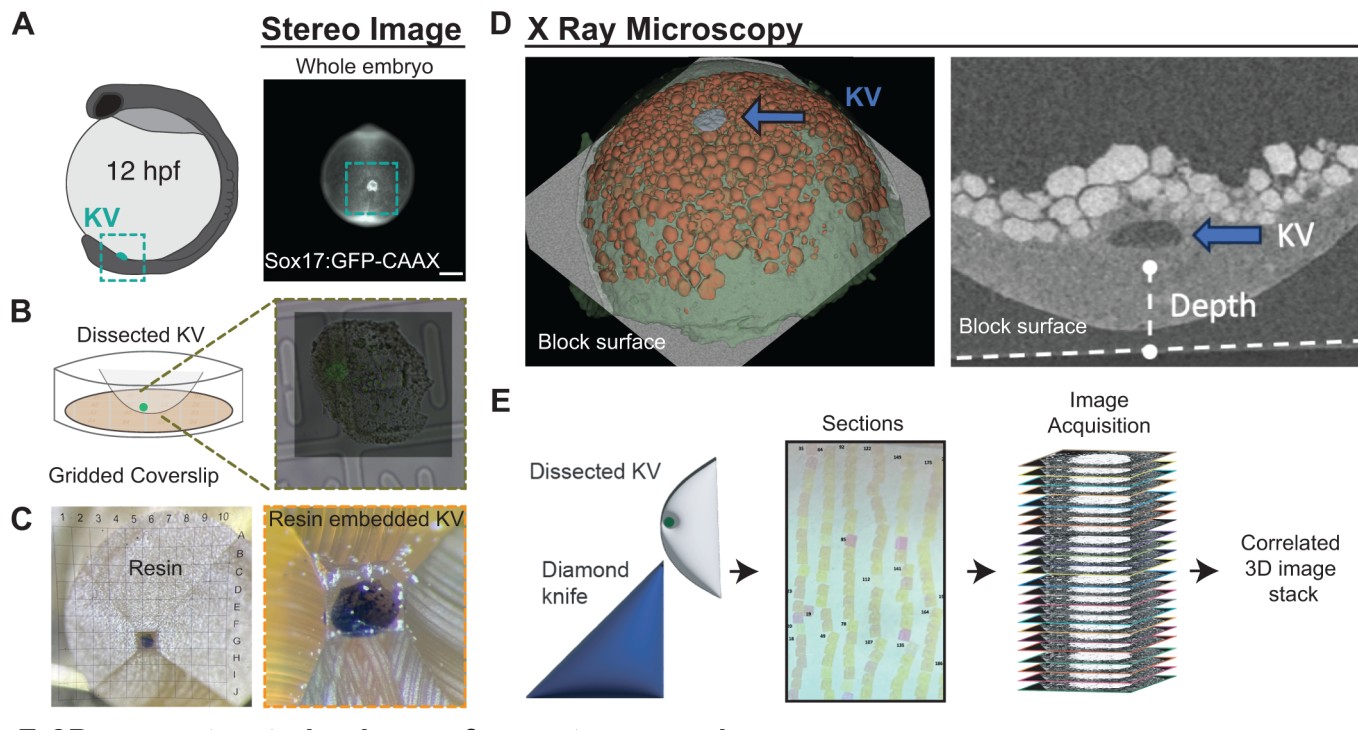

**F** **3D reconstructed volume of array tomography**

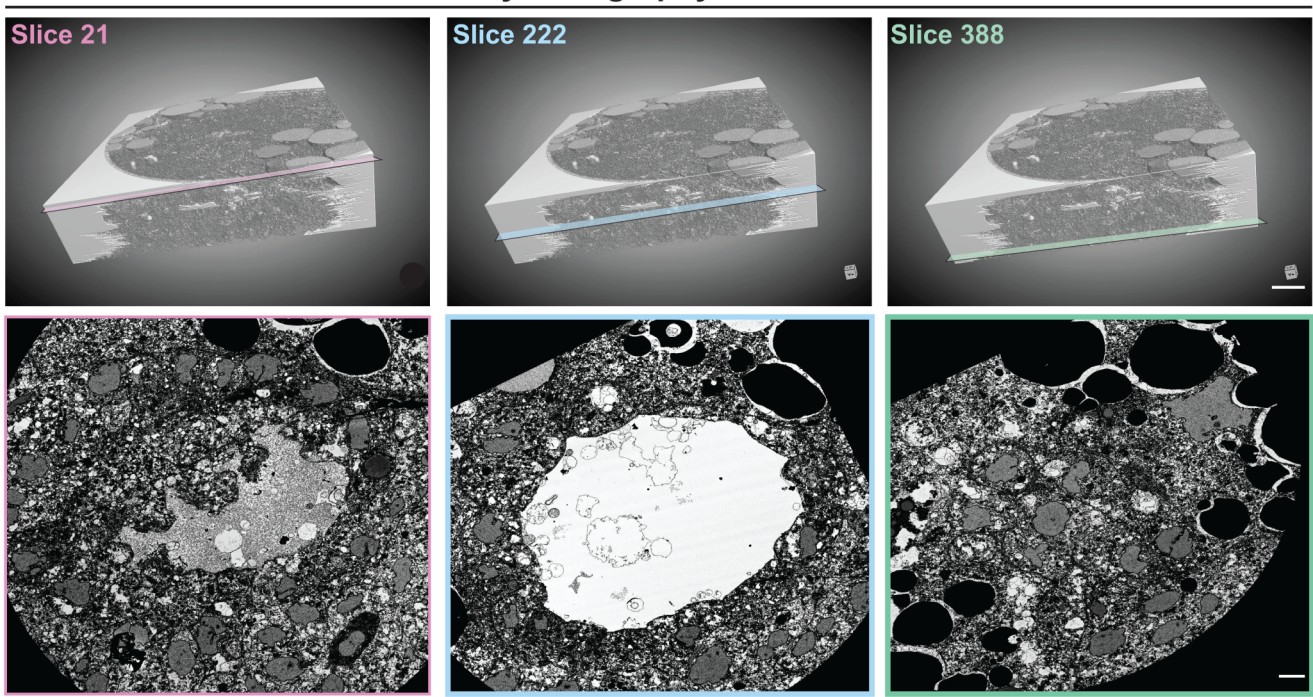

**Fig. 2. Workflow overview for vEM imaging of the KV.** (A) Model of a zebrafish 12 hpf embryo highlighting the KV (cyan, left), alongside an image of a real 12 hpf embryo with Sox17:GFP-CAAX-labeled KV (cyan box) acquired by stereomicroscopy (gray). Scale bar: 2 µm. (B) Schematic of dissected KV mounted on a gridded coverslip; inset shows corresponding microscope image. (C) Resin-embedded KV with gridded scale; zoomed image shows KV within the resin block. (D) Left: X-ray microscopy image of the resin block with segmented annotations for skin (green), yolk cells (orange), and KV (blue arrow). Right: X-ray microscopy image of the KV (inverted gray). (E) Schematic of ultramicrotomy process; schematic of computational stacking of acquired images. (F) Volumetric reconstruction of the KV with boxes indicating three slice locations; corresponding two-dimensional slices from specified serial sections. Scale bars: 5 µm. Refer to Movie 1.

and enabled spatial mapping of subcellular structures within their native tissue context. Consistent with polarized epithelial organization, nuclei localized basally while cilia projected apically into the lumen (Fig. 3B), validating the fidelity of the segmentation and providing a spatial framework for subsequent analyses.

One analytical strength of vEM is its ability to directly resolve centriole number and ultrastructural identity – features that cannot be definitively assessed by light microscopy alone. In cycling cells, centrosomes typically comprise a pair of centrioles: a mature mother centriole, distinguished by DAs and SDAs and capable of nucleating a

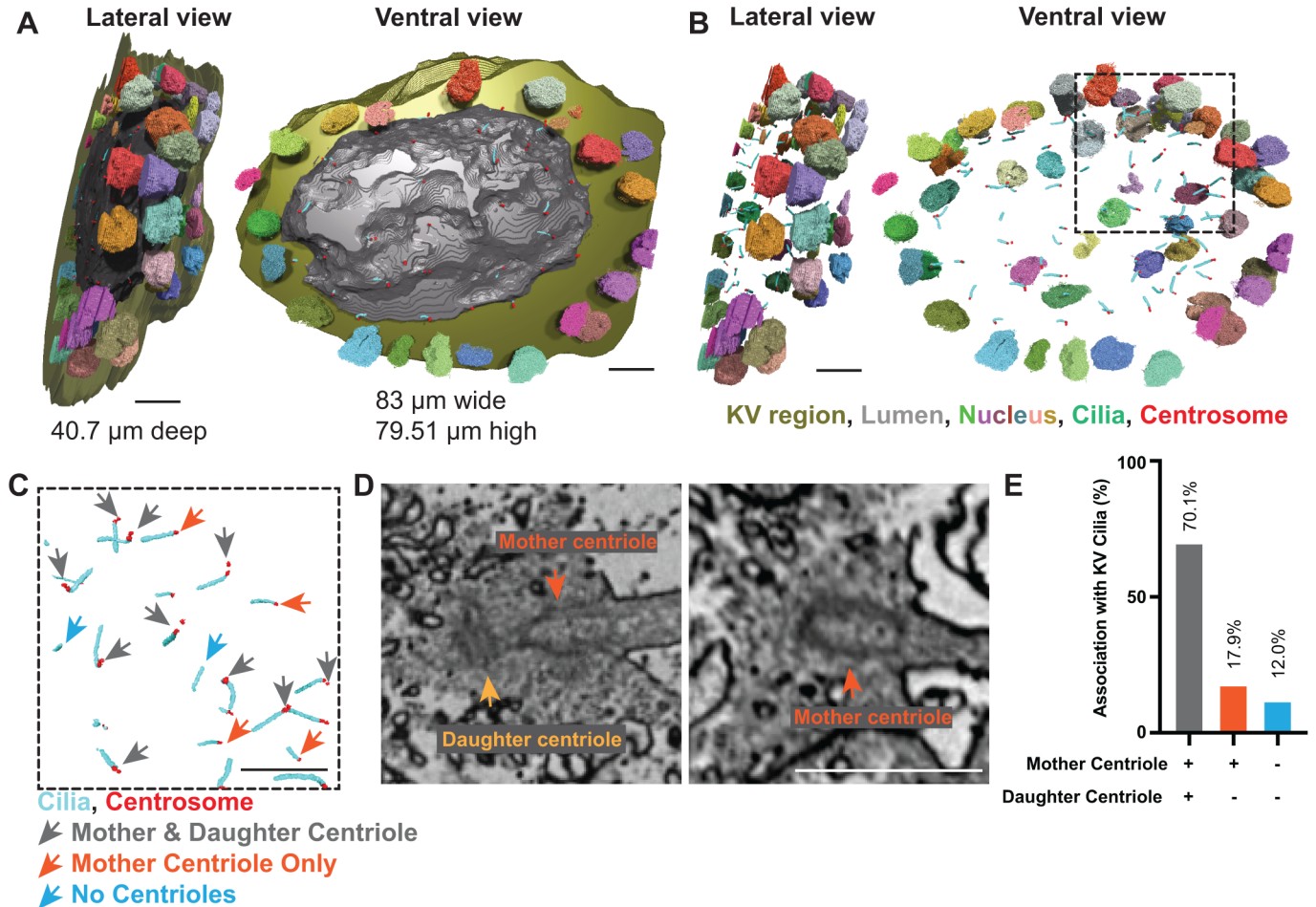

**Fig. 3. vEM of the KV reveals that most cilia associate with both mother and daughter centrioles, but a subset lack at least one.**
(A) Three-dimensional reconstruction of the segmented KV volume showing overall dimensions. Left: lateral view with KV region (green), lumen (gray), nuclei (multi-colored), cilia (cyan), centrosomes (orange), and z-axis measurement (40.7 µm). Right: ventral view with x-axis (83 µm) and y-axis (79.51 µm) measurements. Scale bars: 10 µm. (B,C) Three-dimensional segmentation of KV organelle distribution showing nuclei (multi-colored), cilia (cyan), and centrosomes (orange). Left: lateral view. Right: ventral view. (C, inset) Zoomed view of segmented cilia and centrosomes, showing cilia having mother centriole and daughter centriole (gray arrow), mother centriole only (orange arrow) and no centriole (blue arrow). Scale bars: 5 µm. Refer to Movie 2.
(D) Single two-dimensional slice from the array tomography dataset showing a KV cell with clearly resolved mother and daughter centrioles (left) or mother centriole only (right). Scale bar: 1 µm. (E) Bar graph quantifying the percentage of KV cilia in a single embryo that are associated with no centrioles, only a mother centriole, or both mother and daughter centrioles. Data were obtained from a fully segmented vEM dataset. (n=67 cilia). Refer to Table S1.

cilium, and a daughter centriole that matures in the subsequent cell cycle (Vertii et al., 2016a). In contrast, centriole number can be reduced during terminal differentiation, with documented centriole elimination in germ cells and differentiated somatic tissues, particularly in *Caenorhabditis elegans* and *Drosophila* (Kalbfuss and Gönczy, 2023). Ultrastructural studies have demonstrated that ciliated cells may lose centrioles after initiating axoneme formation, retaining pericentriolar material without an intact centriole barrel (Garbrecht et al., 2021; Magescas et al., 2021; Serwas et al., 2017).

We used vEM to segment and classify centrosomes associated with individual cilia based on the presence of identifiable mother and daughter centrioles (Fig. 3D). We segmented all this KV's cilia (67), and across the 67 cilia segmented within this KV (cyan in Fig. 3B,C), 47 (70.1%) were associated with both a mother and daughter centriole (red segmented centrioles; gray arrows in Fig. 3C), 12 (17.9%) retained only a mother centriole (orange arrowheads in Fig. 3C), and 8 (12.0%) lacked any detectable centriole structure (blue arrowheads indicating cilia without segmented centrioles in Fig. 3C). These distributions are summarized in Fig. 3E, and the full segmentation is shown in Movie 2.

Importantly, this analysis highlights both the strengths and limitations of vEM-derived quantification. Immunofluorescence analysis (Fig. 1) indicated that all KV cilia associate with γ-tubulin-positive centrosomes; however, such markers report centrosome-associated material rather than the integrity of the underlying centriole structure. This limitation is especially relevant in zebrafish, where reliable centriole-specific markers independent of pericentriolar material remain scarce, making it difficult to distinguish intact centrioles from structurally disassembled centrioles by light microscopy alone. By directly resolving centriolar ultrastructure, vEM overcomes this ambiguity and reveals that centrosome-associated material can persist even in the absence of one or both centrioles. Nevertheless, because vEM captures a fixed developmental timepoint, it cannot distinguish permanent centriole elimination from transient remodeling or stage-specific variation.

Whole-organ vEM datasets are also technically demanding and low throughput, limiting the number of embryos that can be analyzed. Accordingly, quantitative distributions derived from single or few volumes should be interpreted as defining the range of structural configurations present, rather than as population-level

frequencies. Developmental heterogeneity, subtle staging differences, and sampling depth represent additional caveats.

Together, these considerations illustrate how vEM provides a high-resolution structural framework for assessing centriole–cilia organization within an intact vertebrate LRO, while emphasizing the need for cautious interpretation. Rather than offering exhaustive statistics, vEM defines what architectures exist and which assumptions based on molecular localization warrant re-evaluation, thereby establishing a foundation for future functional and developmental analyses.

## Structural heterogeneity of mother centrioles associated with KV cilia includes variable appendages and rootlet formation

To extend our analysis of ciliary and centrosome heterogeneity in KV cells, we examined centrosome-associated structures in the vEM dataset, building on earlier observations from Figs 2 and 3. We focused on rootlets and known features of the mother centriole, specifically DAs and SDAs (modeled in Fig. 4A). In most vertebrate cell culture systems, when cells are not ciliated, DAs and SDAs are typically organized with ninefold symmetry (Chong et al., 2020; Kanie et al., 2025; Lau et al., 2012). DAs are critical for docking the mother centriole to the plasma membrane, thereby licensing ciliogenesis (Burke et al., 2014; Joo et al., 2013; Tanos et al., 2013; Ye et al., 2014), while SDAs serve as anchoring sites for microtubules and, in some ciliated cells, are reduced to a single structure known as the basal foot (Delgehyr et al., 2005; Ishikawa et al., 2005; Kunimoto et al., 2012). The basal foot plays a role in orienting the cilium relative to the cell cortex and in coordinating tissue-level processes that depend on directional ciliary beating (reviewed in Dasgupta and Amack, 2016; Hall and Hehnly, 2021; Vertii et al., 2016b). These observations suggest that appendage organization, while often stereotyped, may contribute an additional layer of heterogeneity to centrosome and cilia structure in developing tissues.

During segmentation and analysis, we consistently observed a centrosome-associated dense region (CaDR) surrounding centrioles, which may correspond to the pericentriolar material (PCM) (Doxsey et al., 1994; Fig. 4A), and centrosome-proximal membranes (CPMs) (Fig. 4A), likely representing centriolar vesicles described in previous studies (Hehnly et al., 2012; Knödler et al., 2010; Westlake et al., 2011). The mother centrioles (Fig. 4B) were readily distinguishable from the daughter (both shown in Fig. 3D), with appendages localized exclusively to the mother centriole. CaDRs and CPMs were consistently present near centrioles across all cells examined (Fig. 4B). Rootlet fibers were also visualized (Fig. 4B).

To characterize the prevalence of centrosome-associated features, we analyzed their presence across the entire segmented KV volume (Fig. 4C). CaDRs and CPMs were consistently observed in 100% of cilia. In contrast, centriole appendage composition was more variable: 22.58% of mother centrioles contained both DAs and SDAs, 62.9% had only SDAs, and 9.68% had only DAs. Overall, SDAs were observed more frequently (91.8%) than DAs (33.9%). In some centrioles oriented perfectly, we could observe that DAs had ninefold symmetry (Fig. 4B), whereas SDAs never did (with only a single SDA often present; see Fig. 4B,D). Notably, centrioles that were associated with rootlets (only 5.08%, three cases) always possessed both DAs and an SDA.

In Fig. 4B, we present representative single sections of mother centrioles in which DAs, an SDA, or a rootlet fiber can be clearly visualized. Each image corresponds to a distinct centriole and is intended to illustrate individual centrosomal features rather than multiple structures derived from a single centriole. To demonstrate

the coexistence of multiple features within one centriole, we additionally show representative serial sections from an individual centrosome/basal body (Fig. 4D). This centrosome was rendered in three dimensions; however, the rendering reflects only those DAs that could be confidently resolved given the orientation of the centriole within the volume. Consequently, the number of DAs visualized likely represents a subset of the total present. This analysis highlights that DAs are more challenging to consistently resolve in this vEM datasets, whereas centrioles, SDAs and rootlet fibers are more reliably identified.

The vEM data reveal heterogeneity in centriole-associated architecture, including rootlets. Rootlets, which are classically associated with the mother centriole, were detected in a small subset of KV cells and were structurally resolved in three of 67 mother centrioles (4.5%) (Fig. 4C). This low frequency contrasts with our immunofluorescence data, in which a substantially larger fraction of centrosomes (60.33±23.78%) were positive for the rootlet-associated protein Rootletin (Fig. 1). Together, these observations indicate that while Rootletin can localize to centrosomes, it may not consistently assemble into a morphologically identifiable rootlet structure. Spatial mapping of the vEM dataset further showed that the rootlet-positive cilia detected by ultrastructural criteria were localized to the anterior-left quadrant of the KV (Fig. 4E).

## Ciliary heterogeneity in KV has a predominant cilia subtype exhibiting membrane-associated vesicles and lacking a ciliary pocket

While analyzing cilia in the vEM dataset, we identified cilia with discernible axonemes (modeled in Fig. 5A and shown in Fig. 5B). Motile cilia are classically described as having a 9+2 microtubule arrangement, whereas non-motile (sensory) cilia in vertebrates are primarily 9+0 (Marra et al., 2016). We had hoped to map the distribution of 9+0 and 9+2 cilia across KV cells; however, although some cross-sections (e.g. Fig. 5B) suggest a 9+2 morphology, the resolution of many sections prevented confident assignment of axonemal architecture in all cases.

In contrast, what we could consistently resolve were cilia frequently accompanied by cilia-associated vesicles (CaVs), either fused to or positioned near the axoneme (modeled in Fig. 5A, shown in vEM images in Fig. 5C, bottom, three-dimensional segmentation of a cilium with CaV in Fig. 5F). These vesicles often appeared to be forming from or fusing with the ciliary membrane, a process more clearly visualized by transmission electron microscopy (TEM) (Fig. 5E), suggesting that they may arise directly from the ciliary membrane or represent incoming vesicles from another source. By comparison, ciliary-associated dense vesicles (CaDVs) were observed only in association with the ciliary membrane, without discernible evidence of membrane fusion or pinching events at this resolution (Fig. 5C, top), raising the possibility that these vesicles do not originate from the cilia but rather from an external source.

Ciliary membrane-associated vesicles have been reported in multiple systems (Nager et al., 2017; Nikonorova et al., 2025; Volz et al., 2021; Wang et al., 2021, 2024a, 2013), but to our knowledge have not been described in the LRO. These vesicles are thought to mediate the targeted delivery and removal of membrane proteins, reflecting active remodeling of the ciliary membrane. They have also been implicated in intercellular communication and the compartmentalization of signaling activity (Luxmi and King, 2022; Wang and Barr, 2016; Wang et al., 2024b).

We observed very few cilia displaying a ciliary pocket in this dataset (Fig. 5D,F). The ciliary pocket is a membrane invagination at the base of the cilium implicated in regulating vesicular trafficking,

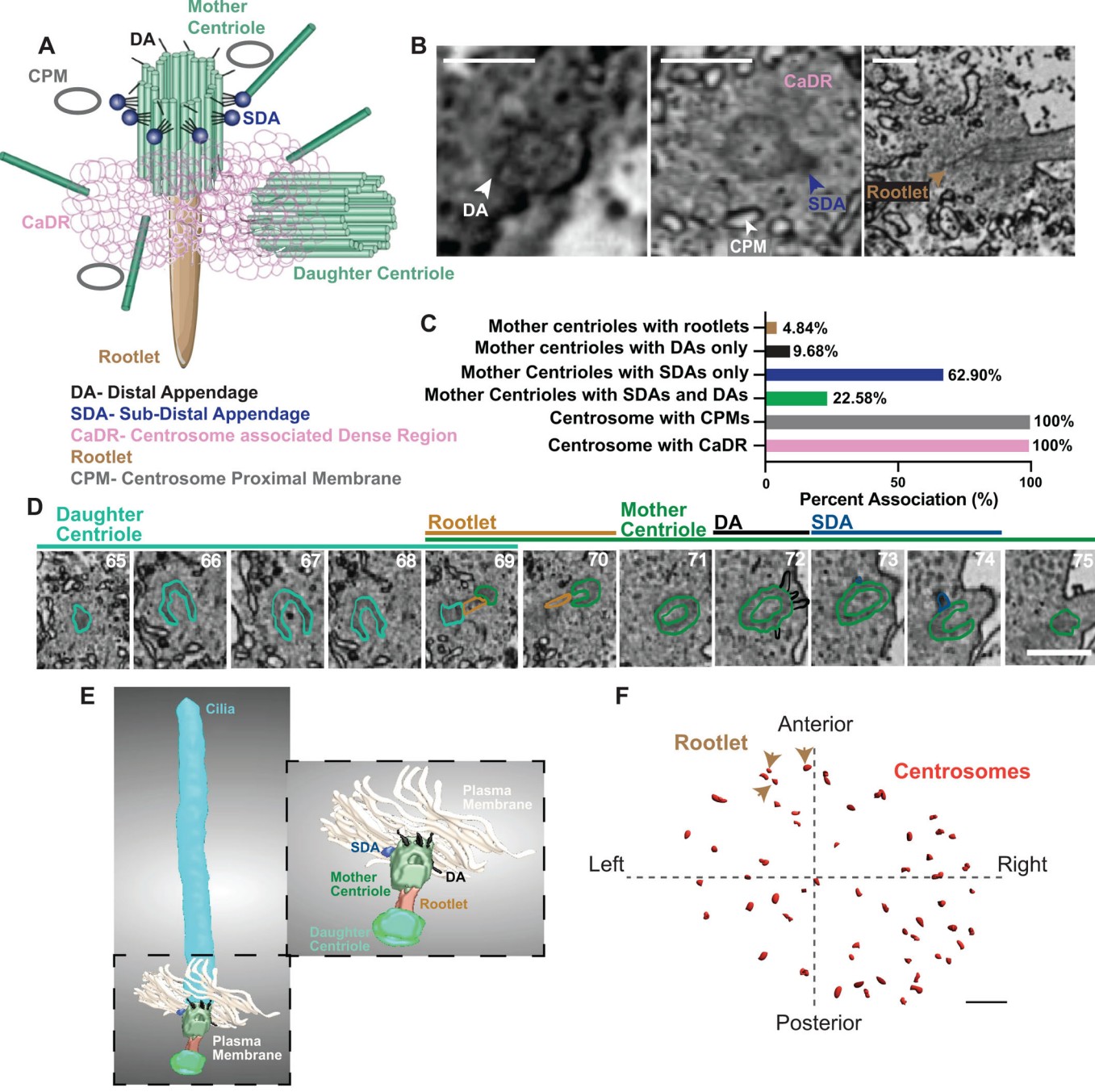

**Fig. 4. Structural heterogeneity of mother centrioles associated with KV cilia includes variable appendages and rootlet formation.** (A) Schematic model of centrosome architecture highlighting the mother and daughter centrioles, distal appendages (DA, black), subdistal appendages (SDA, navy), centrosome-associated dense region (CaDR, pink), centrosome-proximal membrane (CPM, gray), and rootlet fibers (tan). Model adapted from Doxsey (2001). This image is not published under the terms of the CC BY license of this article. For permission to reuse, please see Doxsey (2001). (B) Representative two-dimensional single-slice images from the vEM dataset showing examples of centrosome-associated structures, including DAs, SDAs, CaDRs, CPMs, and rootlet fibers. Colored arrowheads and labels indicate each structure. Scale bars: 0.5 µm. (C) Bar graph quantifying the percentage of centrosomes with each associated structure among KV cells with mother centrioles (for rootlets, DAs, SDAs) or cilia (for CPMs, CaDRs). (*n*=59 mother centrioles). Refer to Table S1. (D) Single two-dimensional slices from the vEM dataset showing a KV mother and daughter centriole with resolved distal and subdistal appendages and a rootlet. Scale bar: 0.5 µm. (E) A three-dimensional segmented mother centriole (green) with distal appendages (DA, black), subdistal appendages (SDA, navy) and rootlet fibers (tan) and a daughter centriole (light green) is shown. (F) Three-dimensional reconstruction of the KV, showing the spatial distribution of centrosomes (red dots) and centrosomes with rootlet fibers (tan arrowheads) along the anterior–posterior axis of the embryo. Scale bar: 5 µm.

selective cargo entry, and protein turnover during ciliogenesis. It is a hallmark feature of many vertebrate cell types and plays critical roles in sensory transduction and signal coordination (Ghossoub et al., 2011; Molla-Herman et al., 2010; Rohatgi and Snell, 2010).

Analysis revealed that most cilia contained CaVs (73.13%). In contrast, only a small subset displayed CaDVs; (2.98%) or invaginated bases (IVBs)/cilia pockets (3.57%) (Fig. 5G). Observations in which the presence or absence of a ciliary pocket

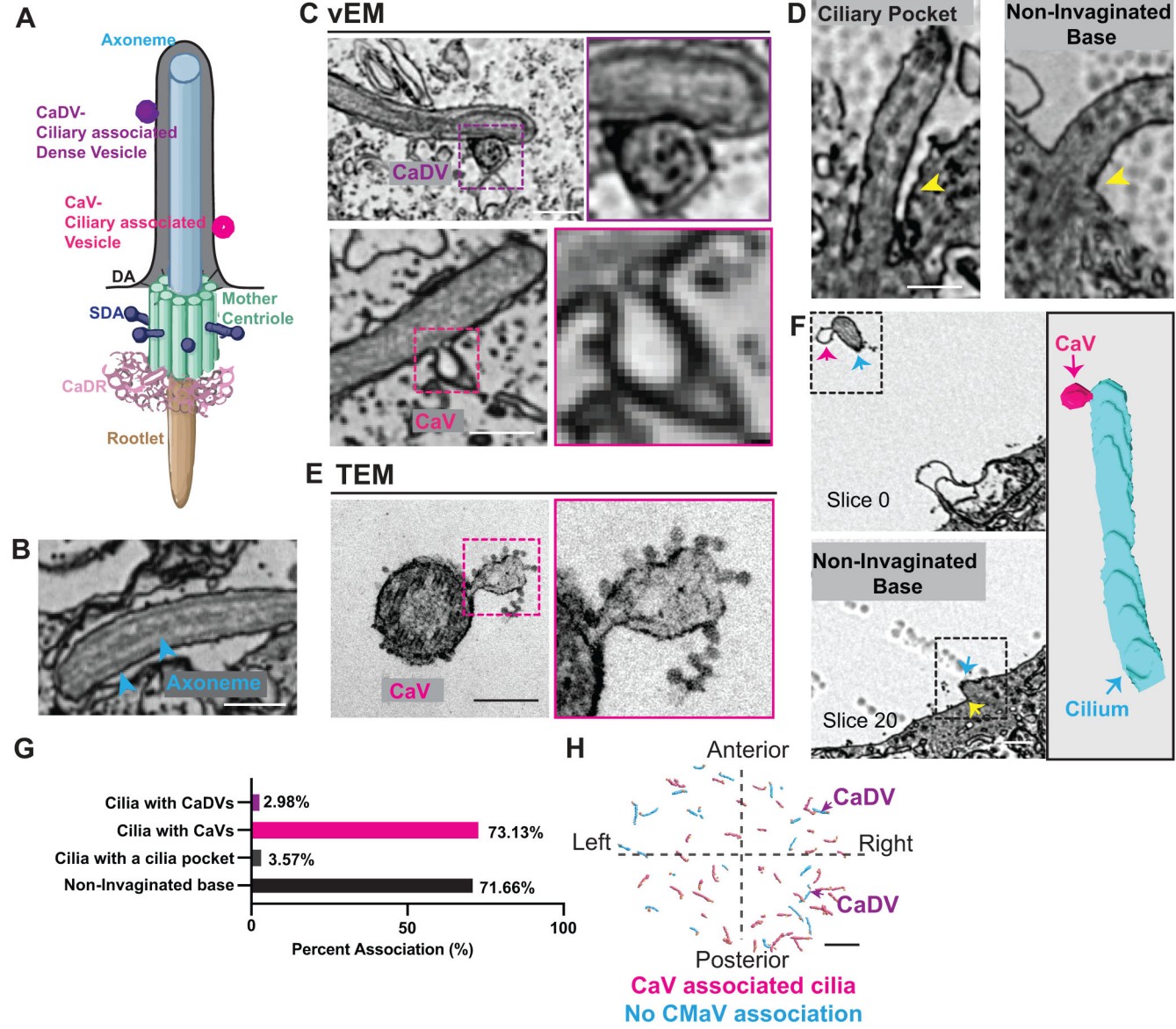

**Fig. 5. Ciliary heterogeneity in KV has a predominant cilia subtype that exhibits membrane-associated vesicles and lacks a ciliary pocket.**
(A) Schematic representation of a cilium with an axoneme surrounded by a membrane bearing two types of associated vesicles: ciliary-associated dense vesicles (CaDV, purple) and ciliary-associated vesicles (CaV, magenta). The mother centriole is depicted with DAs, SDAs, CaDR, and rootlet. (B–D) Representative two-dimensional slices from the serial vEM dataset. (B) Axoneme shown, marked by blue arrowheads. (C) Cilia with CaDV (top) and CaV (bottom); insets show magnified views highlighting differences in vesicle morphology. (D) Examples of invaginated vesicle bases (IVB) and non-invaginated vesicle bases (NIVB), marked by yellow arrowheads. Scale bars: 0.5 µm. (E) Transmission electron micrograph of KV cilia cross-section that contains a ciliary vesicle. Scale bar: 0.2 µm. (F) Representative vEM slices from slice 0 to slice 20 (note the non-invaginated base, marked by yellow arrowhead), used for three-dimensional reconstruction (right). The reconstruction demonstrates a KV cilium (blue, marked by blue arrowheads and arrow) associated with a CaV (magenta arrowhead and arrow). Scale bar: 0.5 µm. (G) Bar graph quantifying the percentage of cilia associated with CaDV, CaV, IVB, and NIVB from the vEM dataset (n=67 cilia from KV dataset). Refer to Table S1. (H) Spatial map of CaV+ (pink) and CaV– (blue) vesicles in KV cells, showing localization relative to the anterior–posterior and left–right axes of the embryo. Cilia associated with CaDV are indicated by purple arrows. Scale bar: 5 µm.

could not be clearly resolved were excluded from the analysis. Spatial mapping showed that CaVs were more frequently observed on the posterior and left sides of the KV, whereas the two cilia that had CaDVs were restricted to the right side (cilia with CaDVs, arrows, Fig. 5H).

## DISCUSSION

Understanding how the vertebrate LRO is assembled requires not only knowledge of molecular pathways and ciliary function, but also a detailed appreciation of the three-dimensional ultrastructural organization of the tissue. In this study, we apply vEM to the

zebrafish KV to generate an ultrastructural map of cilia, centrosomes, and cilia associated vesicles within an intact LRO. Our work highlights how vEM can be leveraged as a platform to define structural heterogeneity, test assumptions derived from light microscopy and identify architectural features that may contribute to LRO function.

A major advantage of vEM is its ability to distinguish molecular localization from ultrastructural assembly. Immunofluorescence approaches have been indispensable for mapping ciliary components in KV cells, yet they inherently report protein presence rather than structural integrity. By contrast, vEM resolves centrioles, appendages,

rootlets, and membrane interfaces directly, allowing structural states to be assessed without reliance on molecular proxies. Applying this approach, we find that while most KV cilia are associated with both mother and daughter centrioles, a substantial fraction retains only a mother centriole or potentially lack detectable centrioles altogether (Fig. 3). Importantly, CaDRs and proximal membranes persist even in varying number of centriole barrels, indicating that centrosome-associated organization can be maintained independently of centriole structure. These observations refine our understanding of centrosome architecture in ciliated tissues and caution against equating centrosome markers with intact centrioles.

Beyond centriole number, vEM enabled analysis of mother centriole appendages and rootlet structures across the entire KV (Fig. 4). We observed marked heterogeneity in DAs, SDAs, and rootlet fibers, with SDAs being the most frequently retained feature and rootlets detected in only a small subset of cells. This graded retention suggests that centrosome remodeling in KV cells may proceed through selective loss of structural elements, rather than wholesale centriole removal. Similar stepwise centriole disassembly has been described in diverse developmental contexts, including sensory neurons and germ cells, supporting the idea that centriole elimination is a regulated and modular process (Kalbfuss and Gönczy, 2023). While our data do not establish the timing or mechanisms underlying this remodeling, they define a spectrum of centrosome architectures present within a single developing organ.

A notable insight from this study is the discrepancy between structural and molecular measures of rootlet presence. Whereas immunofluorescence revealed localization of the rootlet protein Rootletin (Fig. 1), vEM detected assembled rootlet fibers in only a minority of cells (Fig. 4). This finding underscores a central interpretive principle for ultrastructural studies: protein localization does not necessarily imply formation of the corresponding higher-order structure. vEM thus provides an essential corrective lens through which molecular heterogeneity can be reinterpreted in structural terms.

In addition to centrosome architecture, vEM revealed extensive heterogeneity in CaVs (Fig. 5). While limitations in resolution and orientation precluded confident classification of axonemal microtubule arrangements across all cilia, membrane-associated features were consistently resolved. The majority of KV cilia were associated with CaVs, whereas ciliary pockets and CaDVs were rare (Fig. 5).

At the same time, our study highlights important limitations inherent to whole-organ vEM analyses. Acquisition and segmentation of complete volumes are technically demanding and low throughput, constraining the number of embryos that can be analyzed and limiting statistical power. vEM captures a single developmental timepoint, making it difficult to distinguish stable structural states from transient intermediates or to infer temporal sequences of remodeling. In addition, detection of small or orientation-sensitive features – such as DAs or shallow membrane invaginations – may be underestimated in full-volume reconstructions. These considerations emphasize that vEM datasets are best interpreted as defining the range of structural possibilities present within a tissue, rather than providing exhaustive population-level frequencies.

Despite these caveats, the strength of vEM lies in its ability to reveal architectural features that are otherwise inaccessible and to generate testable hypotheses about tissue organization. By providing a three-dimensional ultrastructural framework for the zebrafish KV, this study establishes a reference for future work integrating vEM with live imaging, molecular perturbation, and comparative developmental analyses. More broadly, our findings illustrate how vEM can be applied to ciliated organs to uncover structural heterogeneity and refine interpretations of molecular data.

## MATERIALS AND METHODS

### Zebrafish lines
Zebrafish lines were maintained according to the approved protocols of the Institutional Animal Care Committee of Syracuse University (IACUC Protocol #18-006). Embryos were raised at 28.5°C and staged (as described in Kimmel et al., 1995). Transgenic zebrafish lines used for vEM and immunohistochemistry are listed in Table S2.

### Immunofluorescence
Immunostaining for acetylated tubulin, γ-tubulin, IFT88, Rootletin, and GFP was performed on transgenic zebrafish embryos [*Tg(Sox17:GFP-CAAX)*] fixed at 8, 10, and 12 hpf in 4% paraformaldehyde (PFA) with 0.5% Triton X-100 at 4°C overnight. Embryos were dechorinated after washing with PBST (0.1% Tween-20 in PBS) three times. Embryos were blocked in wash solution (1%DMSO, 1% BSA,0.1% Triton X-100) for 4 h at room temperature with gentle agitation. Primary antibody incubation (diluted in wash solution) occurs overnight at 4°C. Primary antibodies used include the following: anti-IFT88 (rabbit) (1:200, Proteintech, 13967-1-AP: AB_2121979), anti-γ-Tubulin (goat) (1:200, Santa Cruz Biotechnology, sc-7396), anti-GFP (chicken) (1:300, GeneTex, GTX13970: AB_371416), anti-acetylated Tubulin (mouse) (1:300, Sigma-Aldrich, T6793: RRID: AB_477585), anti-GFP (rabbit) (1:300, Molecular Probes, A-11122: AB_221569), and anti-Rootletin (chicken) (1:300, Fisher Scientific, ABN1686MI), refer to Table S2. Embryos were washed, blocked for an hour, and incubated with secondary antibodies for 2-4 h at room temperature or overnight at 4°C. Secondary antibodies used include the following: Alexa Fluor Anti-Mouse 568 (1:300, Life Technologies, A10037; RRID: AB_2534013), Alexa Fluor Anti-Chicken 488 (1:300, Fisher Scientific, A11039), or Alexa Fluor Anti-Mouse 647 (1:300, Life Technologies, A31571; RRID: AB_162542), Alexa Fluor Anti-Rabbit 568 (1:300, Life Technologies, A21206; RRID: AB_2535792), Alexa Fluor Anti-Rabbit 647 (1:300, Fisher Scientific, A31573; RRID: AB_2536183), Alexa Fluor Anti-Goat 647 (1:300, Jackson ImmunoResearch, 705-605-003; RRID: AB_2340436). Embryos were stained with DAPI (1 mg/ml) to label nuclei after washing three times with wash solution. Embryos were mounted with 2% agarose after washing with PBS. Refer to Table S2.

### Imaging
The Leica SP8 laser scanning confocal microscope (LSCM) and the Leica DMi8 with a spinning disk confocal were used to image KV development and lumen formation in zebrafish embryos. The SP8 system was equipped with HC PL APO 20×/0.75 IMM CORR, HC PL APO 40×/1.10 W CORR water, and HC PL APO 63×/1.3 Glyc CORR glycerol objectives, with image acquisition performed using LAS-X software. The DMi8 spinning disk system included an X-light V2 confocal unit, Visitron VisiFRAP-DC photokinetics (405 and 355 nm lasers), a Lumencore SPECTRA X light source, a Photometrics Prime-95B sCMOS camera, and an 89 North-LDi laser launch. Objectives used with this system included HC PL APO 40×/1.10 W CORR water, HC PL APO 40×/0.95 NA CORR dry, and HCX PL APO 63×/1.40–0.60 NA oil. Images were captured using VisiView software. For staging, a Leica M165 FC stereomicroscope equipped with a DFC 9000 GT sCMOS camera was used.

### Association analysis
Association between Ac-tubulin and γ-tubulin, Ac-tubulin and IFT88, and Rootletin and γ-tubulin signals was assessed by manually counting the number of cilia or centrosomes exhibiting overlapping or adjacent immunofluorescent staining as described in the text. The percentage of cilia-positive structures associated with centrosomes (γ-tubulin) or IFT88-positive cilia was calculated for each KV by dividing the number of associated cilia by the total number of cilia counted. Similarly, the percentage of γ-tubulin-positive centrosomes associated with Rootletin was obtained by dividing the number of colocalized centrosomes by the total number of centrosomes counted.

## vEM sample preparation

Zebrafish embryos at 12 hpf were fixed in freshly prepared 4% PFA. The KV region was dissected and deyolked in 4% PFA, then oriented on a gridded Mattek dish using a fluorescence stereoscope. Post-fixation was performed in Karnovsky's fixative (2.5% glutaraldehyde, 2% formaldehyde in 0.1 M sodium cacodylate, pH 7.4) for 2 h at room temperature, followed by five 3-min washes in 0.1 M sodium cacodylate buffer (pH 7.4). Samples were stored at 4°C until further processing. Samples were rendered stationary for subsequent steps by dropping a small but sufficient volume of 1% low-melt agarose solution directly on the gridded Mattek plate to embed the sample *in situ*. Samples were then incubated in 2% aqueous osmium tetroxide and 1.5% potassium ferricyanide for 1 h, then rinsed in ddH$_2$O until the agarose was visibly clear. Samples were stained overnight at 4°C with 1% aqueous uranyl acetate and rinsed five times in ddH$_2$O. Lead aspartate staining was performed by incubating samples in a solution of 0.066 g lead nitrate in 10 ml of 0.03 M aspartic acid (pH adjusted to 5.5 with 1 M KOH) at 60°C for 30 min, followed by thorough rinsing in ddH$_2$O. Dehydration was carried out using a graded ethanol series (35%, 50%, 70%, 95%, and 100%),10 min at each step. Dehydration helped solidify the agarose-embedded sample to the point that it could be carefully moved into a glass vial. After this, samples were washed thrice for 10 min each in propylene oxide. Tissue infiltration was achieved with Polybed 812 resin using a graded sequence: 1:3 resin: propylene oxide for 1 h, 1:1 resin: propylene oxide overnight, 3:1 resin: propylene oxide for 5 h, and, finally, 100% resin overnight. The agarose-embedded embryos were transferred to embedding molds containing freshly degassed resin and polymerized at 60–65°C for 48 h. Embedded tissue blocks were trimmed into a pyramid shape, and the position of the KV was verified using X-ray microscopy (Zeiss/Xradia Versa 730). The FIJI plugin Crosshair was used to target the precise angle and depth of the KV in the resin block based on the X-ray microscopy data (Meechan et al., 2022). Ultrathin serial sections (~70 nm) were cut with a diamond knife (Diatome) onto an ITO coverslip (SPI supplies) using a Leica ARTOS ultramicrotome. Coverslips were dried flat on a slide warmer set to ~55°C for 30 min to ensure section adherence and drying.

## vEM image acquisition and reconstruction

Coverslips were mounted onto a 4-inch type-p silicon wafer (EMS) and grounded with conductive copper tape (EMS). Wafers were affixed to a 4-inch stage-decel holder in Ziess SEM (GeminiSEM 450; Carl Ziess) and imaged using ATLAS 5 vEM software (Fibics). Two ITO coverslips were imaged using a four-quadrant backscatter detector, with electron beam operated at 3.2 kV EHT with 2 kV beam deceleration and 600 pA probe current. Low-resolution overview scans were collected at 3000 nm pixel resolution, medium-resolution section sets were collected at 150 nm pixel resolution, and high-resolution sites were collected at 10 nm pixel resolution. Once image acquisition was complete, the image stack was locally cropped and aligned using the ATLAS 5 software. The resulting image stack was exported and then processed using python-based scripts to produce an aligned and contrast/brightness adjusted image dataset: a stack of images at 10 nm (xy)×100 nm (z). Higher-resolution imaging of specific cilia was performed on the same ITO coverslips. The re-imaged areas were collected on the Ziess SEM using a four-quadrant backscatter detector, with electron beam operated at 3.5 kV EHT with 2 kV beam deceleration and 600 pA probes current. High resolution sites of targeted cilia were collected at 3 nm xy pixel resolution and exported. Original imaging notes were used to identify and re-image regions of interest.

## Manual segmentation in dragonfly software

Three-dimensional segmentation of KV structures was performed manually using Dragonfly software. Image stacks, acquired from vEM, were imported into Dragonfly to create a volumetric dataset. Each structure of interest was then manually traced across serial sections using the software's segmentation tools. After each structure's three-dimensional model was generated, a contour mesh was applied with a single sampling (for x, y, and z coordinates) and a threshold of 50. Each structure was smoothed using two to four iterations, with the number of iterations adjusted until pixelated edge artifacts were eliminated, as confirmed by visual inspection. The final reconstructed volume was generated by aligning individual segmented axis providing a detailed three-dimensional representation of the sample.

## Analysis

PRISM9 software was used for all graph preparations that include all individual data points across embryos and clutches denoted by color and size of points, respectively, and as noted in legends. These plots were presented as a bar graph with superplot to denote individual embryos across clutches (Lord et al., 2020).

### Acknowledgements

We thank Jesse Aaron and the Advanced Imaging Center, Howard Hughes Medical Institute Janelia Research Campus for access to the Zeiss Versa XRM instrument for sample location within the resin block for subsequent vEM; and Benjamin Zink at SUNY Upstate Medical University for performing the TEM imaging as a paid service.

### Competing interests

The authors declare no competing or financial interests.

### Author contributions

Conceptualization: F.O., K.N.; Data curation: F.O., A.L.A., M.M., C.D., A.W.S.; Formal analysis: F.O., A.L.A., A.W.S., H.H.; Funding acquisition: K.N., H.H.; Investigation: F.O., A.L.A., A.W.S., H.H.; Methodology: F.O., A.L.A., M.M., C.D., A.W.S., K.N., H.H.; Project administration: F.O., K.N., H.H.; Resources: F.O., K.N., H.H.; Supervision: K.N., H.H.; Validation: A.L.A., H.H.; Visualization: A.L.A., K.N., H.H.; Writing – original draft: F.O.; Writing – review & editing: F.O., A.L.A., K.N., H.H.

### Funding

This work was supported by National Institute of General Medical Sciences (R01GM-127621, R01GM-130874 and R35GM158119 to H.H.), and in part by National Cancer Institute, under Contract No. 75N91019D00024. The content of this publication does not necessarily reflect the views or policies of the Department of Health and Human Services, nor does mention of trade names, commercial products, or organizations imply endorsement by the US Government. Open Access funding provided by Syracuse University. Deposited in PMC for immediate release.

### Data and resource availability

vEM datasets are available at Dryad (https://datadryad.org/dataset/doi:10.5061/dryad.8kprr4z2z). All other relevant data and details of resources can be found within the article and its supplementary information.

### First Person

This article has an associated First Person interview with the first author of the paper.

### Peer review history

The peer review history is available online at https://journals.biologists.com/bio/lookup/doi/10.1242/bio.062489.reviewer-comments.pdf

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
