## [Peer Review File · Biology Open]

Insights into the zebrafish Left-Right Organizer's Centrosomes and Cilia via Volume Electron Microscopy

Favour Ononiwu, Albert Lawrence Adhya, Melissa Mikolaj, Christopher Dell, Abdalla Wael Shamil, Kedar Narayan and Heidi Hehnly

DOI: 10.1242/bio.062489

Editor: Tristan Rodríguez

Review timeline

Submission to sister journal: 18 September 2025

Editorial decision at sister journal: 16 December 2025

Transfer to Biology Open: 3 February 2026

Accepted: 4 February 2026

Original submission to sister journal

First decision letter

MS Title: Insights into the zebrafish Left-Right Organizer's Centrosomes and Cilia via Volume Electron Microscopy

Authors: Favour Ononiwu, Albert Lawrence Adhya, Melissa Mikolaj, Christopher Dell, Abdalla Wael Shamil, Kedar Narayan and Heidi Hehnly

I have now received all the referees reports on the above manuscript, and have reached a decision. The referees' comments are appended below.

As you will see from their reports, the referees recognise the potential of your work, but they also raise significant concerns about it. Given the nature of these concerns, I am afraid I have little choice other than to reject the paper at this stage.

However, having evaluated the paper, I do recognise the potential importance of this work. I would therefore be prepared to consider as a new submission an extension of this study that contains new experiments, data and discussions and that address fully the major concerns of the referees. The work required goes beyond a standard revision of the paper. Please bear in mind that the referees (who may be different from the present reviewers) will assess the novelty of your work in the context of all previous publications, including those published between now and the time of resubmission.

Please ensure that you indicate that this is a resubmission, and enter your manuscript identification number. I would also ask you to provide in the cover letter an explanation of the key ways in which the manuscript differs from the current submission, followed by a point-by-point response to the referees' concerns.

We do understand that the work entailed in a potential new submission is significant, and that you may prefer to submit elsewhere without further delay. Please do let us know if you decide not to resubmit to us, so that we can close our file.

Reviewer 1: SUMMARY OF THE ADVANCE MADE IN THIS PAPER AND ITS POTENTIAL SIGNIFICANCE TO THE FIELD

The report by Ononiwu and colleagues reports on a role for IFT88 in KV morphogenesis along with analysis of a volumetric EM series of a single KV. In the latter, the researchers find that some cells appear to lack daughter centrioles or both mother and daughter centrioles. They also quantify how many centrioles with distal or subdistal appendages they can identify. Finally, they quantify how many axonemes are associated with vesicles and with ciliary pockets.

Overall, it is clear that a lot of work has been done and is reported on here. As a reviewer I appreciate the clarity of the text and thoughtful figures. The optogenetic sequestration of IFT88 is an interesting tool and the effect on KV morphogenesis is unexpected. Having the volumetric EM series is intriguing. However, there are too many issues with the interpretation of the data for me to be able to support publication in its current form.

SUGGESTIONS TO AUTHORS

Major issues:

1. Loss of cilia has not previously been reported to affect KV formation as dramatically as is seen here with IFT88 sequestration. Loss of *foxj1* does not affect KV lumen formation for example. This would suggest that the effect the authors are observing is due to IFT88 and not to cilia.

On this point Line 120-122 - It would be more accurate to say the results support a model where IFT88 plays an active, cell autonomous role in KV morphogenesis.

2. I recognize how difficult this dataset was to obtain and analyze but given how variable KV sizes and cell numbers can be, it is possible this particular KV is an outlier, and thus conclusions drawn have to be taken with healthy skepticism. Especially when the conclusions are based on two or three instances in a large dataset. In the S1 movie, not all slices appear to be included, which may be necessary for creating a reasonably sized movie. But how many slices were unable to be processed or imaged? Is it possible centrioles were missed because of this? Are they confident that not being able to visualize something equates with absence and not the orientation? The authors do mention that they were unable to accurately visualize axonemes to make conclusions about 9+0 or 9+2 due to orientation issues. How confident are they that they were able to accurately determine how many centrioles had distal and subdistal appendages? The conclusions being made are too strong given the potential caveats to their analysis.

3. It is very premature to assume something can be learned about left-right patterning from the data presented here. Given how many issues there may be with the EM analysis, the focus on left-right patterning is unwarranted.

Other points:

Line 84 - does the system induce clustering at a specific location (eg. Mitochondria)? Or just generally cytoplasmic? Please clarify.

Line 114-116 - How do you know these clusters are sites of shortened cilia? I can't see that from the Fig. 1C. They appear to be either cilia spots (cyan) or CIB1RFP spots. Explain how this conclusion was reached or remove this as a conclusion.

Figure 2 - is it possible to show membranes in these images to get a sense of whether IFT88 is at the tip or base? And whether or not there are multiple gamma tubulin foci in cells?

Line 198 - "and KV (blue)" should be "and KV (blue arrow)".

Figure 5A - CPM is listed as Central Proximal Vesicles, should be membranes.

Line 278 - "The mother and daughter centrioles were readily distinguishable (Figure 5B)." There do not appear to be two centrioles in any of these panels, so the callout is odd. Did you assign the mother centriole label to centrioles with appendages? Or did you ensure all centrioles labeled "mother centrioles" were associated with a primary cilium and then assess the presence or absence of appendages?

Figure 5D - I can't see what is being called a DA or SDA in these two images. Based on the rendering, at least one DA is in the region of SDAs. Is this an orientation issue? Perhaps duplicating the images and coloring the structure would be helpful in addition to the rendering.

Line 310 - with a dataset of 1 - deciding that 3 centrosomes with rootlet fibers that were detectable are asymmetrically localized is a big stretch. This needs more caveats (we may have missed them, need more datasets to be sure, etc) and less emphasis as an impactful finding.

Line 360 - similar concern here. Finding 2 cilia with CADVs doesn't mean they weren't missed in other regions so it's premature to say these are regionalized. Similarly, it does appear that more CaVs were detected in posterior parts of KV, but there isn't a clear L/R bias in the anterior. While this would fit current models about LRO function, I don't think the data here is strong enough to support this conclusion.

Reviewer 2: SUMMARY OF THE ADVANCE MADE IN THIS PAPER AND ITS POTENTIAL SIGNIFICANCE TO THE FIELD

The manuscript combines KV specific optogenetic IFT88 clustering system with high resolution volume electron microscopy (vEM) to interrogate ciliary function and ultrastructure in the zebrafish left right organizer. This approach yields a novel, three dimensional map of centriolar and membrane specializations (e.g., distal/sub distal appendages, rootlets, ciliary associated vesicles) and directly links IFT88 dependent ciliogenesis to lumen morphogenesis.

Interest for the field: Left right (LR) asymmetry relies on cilia driven fluid flow, yet the contribution of cilia to the early architecture of the LR organizer remains poorly defined. By demonstrating that IFT88 disruption abolishes KV cilia and reduces lumen size, the work clarifies a cell autonomous, structural role for cilia in organogenesis. The discovery of marked structural heterogeneity: 70 % of cilia retain both mother and daughter centrioles, while 30 % lack one or both, and the identification of specific vesicle populations adds a new layer of spatial regulation to LR patterning.

The study provides (i) a causal link between IFT88 mediated ciliogenesis and KV lumen formation and (ii) evidence that distinct ciliary associated vesicle subtypes may participate in asymmetric signaling. Together, these findings advance our understanding of how ciliary specialization orchestrates LR axis specification. O

SUGGESTIONS TO AUTHORS

Even though interesting, the study remains superficial and is unclear in its focus and aims. To increase the visibility of the study for the community of developmental biologist, here are a few comments:

1. About the optogenetic method

* The CRY2/CIB1 IFT88 optogenetic system produces a statistically significant drop in the proportion of ciliated KV cells ($\approx 50\%$ reduction) and in cilium length, together with a marked decrease in lumen area ($p < 0.0001$). Only two experimental conditions (global vs KV specific clustering) are shown; additional controls such as a non binding CRY2 mutant or CIB1 without IFT88 would strengthen the claim of specificity and CRY2/CIB1 IFT88 without 488nm illumination.

* The optogenetic system primarily traps the CIB1-RFP-IFT88 fusion protein, but because IFT88 functions as part of a multi-protein IFT-B complex, clustering the fusion protein also pulls endogenous IFT88 (and other IFT-B components) out of circulation. What is the proportion of endogenous IFT88 that is trapped. How does this treatment affect the non-ciliary function of IFT88? Can the authors show IFT88 stainings in these conditions?

* Does the optogenetic treatment lead to left right defects? This should be documented.

* What is the role of IFT88 in lumen formation? There is extensive literature addressing the mechanism of KV morphogenesis, including lumen formation.

2. About the vEM analysis

- * The vEM analysis is based on a single fully segmented KV (67 cilia), which limits the generality of the reported centriole heterogeneity (70.1 % with both centrioles, 12 % lacking centrioles). Replicating the segmentation on multiple embryos and different stages is necessary. It would increase confidence and would constitute a great resource for the community.
- * The presence of three cilia sub-populations—those with both centrioles, only a mother centriole, or no detectable centriole (~12% of cilia)—is interpreted as evidence for progressive centriole elimination during KV maturation. However, this conclusion is based on a single time point (12 hpf) and specimen, making it difficult to determine whether the observed heterogeneity reflects a developmental sequence or intrinsic cell-to-cell variability. To solidify the elimination model, additional experiments such as immunostaining for PCM markers and temporal analysis across different developmental stages are needed to track centriole and appendage dynamics.
- * The functional relevance of the two vesicle subtypes (CaVs enriched posterior left, CaDVs confined to the right) is interesting, yet no statistics is presented. Is there a specific accumulation of these CaVs in asymmetric localisation of the KV. This needs to be addressed to enlarge the scope of the study.
- * The overall analysis should take into account the left-right function of the KV and carefully analyse evidences of asymmetries in its structure.
- * Cite earlier work on distal and sub distal appendages when discussing the new appendage statistics.
- * Deposit the vEM dataset and segmentation scripts in an open repository to enhance reproducibility.
- * Provide full statistical details (sample size per clutch, exact tests) in Table S1.
- * Giving the stage how the embryo in hours is not always robust as a stage indication (it fluctuates with temperature and sometimes with the ZF background). How many somites do the embryo display at this stage in the lab condition.

Minor comment:

- * Please explain better the optogenetic approach, namely that it traps the endogenous IFT88 to the cytoplasm and limits its activity in cilia.

Transfer to Biology Open

Author response to reviewers' comments

We thank the reviewers for their thoughtful and detailed evaluations. In response to the concerns raised, we have substantially reframed the manuscript as a methods- and resource- focused study centered on volumetric electron microscopy (vEM). Interpretive claims related to cilia-dependent morphogenesis and left-right (LR) patterning have been removed and ideas around centriole elimination caveated. Below, we respond point-by-point.

Reviewer 1

“Overall, it is clear that a lot of work has been done and is reported on here. As a reviewer I appreciate the clarity of the text and thoughtful figures. The optogenetic sequestration of IFT88 is an interesting tool and the effect on KV morphogenesis is unexpected. Having the volumetric EM series is intriguing. However, there are too many issues with the interpretation of the data for me to be able to support publication in its current form.”

The manuscript has been refocused to emphasize vEM acquisition, segmentation, and quantitative annotation rather than biological mechanism. Interpretive conclusions extending beyond what can be directly visualized in a single vEM dataset have been removed and/or caveats explicitly explained.

Major Issues:

1. *“Loss of cilia has not previously been reported to affect KV formation as dramatically as is seen here... This would suggest that the effect the authors are observing is due to IFT88 and not to cilia.”*

Determining whether the effects observed here reflect an IFT88-specific requirement or a broader consequence of altered ciliogenesis would require additional experimental validation beyond the scope of the current work. To avoid overinterpretation, we have therefore removed this dataset and refocused the manuscript on the vEM-based methodological framework for ultrastructural analysis of the KV.

2. *“It is possible this particular KV is an outlier... conclusions drawn have to be taken with healthy skepticism.”*

We agree that this could always be a possibility. To mitigate this possibility, we carefully screened embryos for KVs containing approximately 50-70 cells, a range previously reported to support robust left-right symmetry, and exhibiting fully formed lumens prior to vEM processing. We now explicitly discuss these selection criteria and their limitations in the revised text (pg 10-11). The manuscript has also been revised to frame the vEM analysis as a single, fully reconstructed KV, and all language implying generality has been removed. Accordingly, quantitative measurements are presented as descriptive observations rather than population-level conclusions.

“Is it possible centrioles were missed because of this? Are they confident that not being able to visualize something equates with absence...?”

We fully understand this concern. In the revised manuscript, we explicitly state that failure to detect a structure does not necessarily imply its absence, and we address limitations related to orientation, section loss, and segmentation in a dedicated subsection. At the same time, we note that vEM provides a more direct and reliable assessment of ultrastructural presence than immunostaining alone, and we discuss the complementary strengths and limitations of these approaches in the text.

“How confident are they that they were able to accurately determine how many centrioles had distal and subdistal appendages?”

Centrioles are relatively large ultrastructural features, and in cases where a mother centriole could be identified but no daughter centriole was detected, we are reasonably confident in this assessment. Similarly, rootlet fibers, when present, were readily identifiable by their characteristic morphology; conversely, in cases where a mother centriole was resolved but no rootlet was observed, its absence was unambiguous. Subdistal appendages followed a comparable pattern, typically appearing as a single prominent structure with substantial length and width that was readily distinguishable when present.

In contrast, distal appendages were considerably smaller and more sensitive to centriole orientation, which limited our ability to reliably enumerate them. Their detection was therefore more variable, like our ability to resolve the central microtubule doublet of the axoneme. Accordingly, we have softened all related claims and clarified the morphological criteria used, emphasizing that distal appendage identification is orientation-dependent and subject to under-detection.

One factor that underlies our confidence, and which is addressed further below, is the three-dimensional segmentation pipeline used to identify these structures. All segmentations were performed across the complete series of serial 70 nm sections spanning the entire KV volume, enabling reconstruction and analysis in three dimensions. Our analysis followed a stepwise workflow in which we first segmented all KV cell nuclei and identified which cells were ciliated, then segmented all cilia, and finally assessed the presence of mother and daughter centrioles within those cells by segmenting all identifiable centrioles. The outcomes of this workflow are illustrated in Figures 3B-C and Supplemental Video 2, and the description of this process has been clarified in the revised text. (Page 11-14)

3. *“It is very premature to assume something can be learned about left-right patterning from the data presented here.”*

We agree. All conclusions relating to left-right patterning, asymmetric signaling, or flow-dependent mechanisms have been removed or added to the discussion as future areas of inquiry. Spatial observations are reported without functional interpretation.

Other Points

“How do you know these clusters are sites of shortened cilia?”

The optogenetic studies and interpretation has been removed.

“Is it possible to show membranes... whether IFT88 is at the tip or base?”

The IFT88 studies have been removed to expand for future studies.

“The mother and daughter centrioles were readily distinguishable (Figure 5B). There do not appear to be two centrioles in any of these panels, so the callout is odd. Did you assign the mother centriole label to centrioles with appendages? Or did you ensure all centrioles labeled “mother centrioles” were associated with a primary cilium and then assess the presence or absence of appendages?”

The callout refers to the updated Figures 3C-E. In the left panel, two centrioles are resolved and labeled as mother and daughter from an example section, whereas in the right panel only a single mother centriole is detected from a single section. In both examples, distal appendages are visible, and in the left panel a rootlet fiber is also apparent. Serial sections above and below this plane confirm the presence or absence of a daughter centriole (a full serial section example is now included in updated Figure 4D). Centrioles are rendered in Figure 3C (red) from a subset of the full volume shown in Figure 3B, allowing us to account for cases in which mother and daughter centrioles are not captured within the same individual section but are resolved through three-dimensional rendering.

In the updated Figure 4B, we present representative single sections of mother centrioles in which distal appendages, subdistal appendages, or a rootlet can be visualized. Each image corresponds to a distinct centriole, rather than multiple features derived from a single structure.

To illustrate the presence of multiple features within a single centriole, we additionally include representative serial sections from an individual centriole in which both distal and subdistal appendages can be identified (updated Figure 4D). This centriole was rendered in three dimensions, and the rendering reflects only those appendages that could be confidently resolved given the orientation of the centriole. As a result, the rendered distal appendages likely represent a subset of the total complement. This example highlights that distal appendages are more difficult to consistently resolve, whereas subdistal appendages and rootlet fibers are more reliably identified in this dataset. We now clarify this in the text, page 15.

Our analysis followed a stepwise segmentation workflow: we first segmented all KV cell nuclei and determined which cells were ciliated, then segmented all cilia, and subsequently assessed the presence of mother and daughter centrioles within those cells, segmenting all identifiable centrioles. Outcomes of this workflow is illustrated in Figures 3B-C and Supplemental Video 2 and has been clarified in the revised text, pages 10-13.

This approach enabled systematic inspection of each segmented object to identify centrosomal features (appendages and rootlets) as well as ciliary-associated membrane structures, including vesicle subtypes.

“deciding that 3 centrosomes with rootlet fibers that were detectable are asymmetrically localized is a big stretch. This needs more caveats (we may have missed them, need more datasets to be sure, etc) and less emphasis as an impactful finding.”

Within this dataset, only three centrosomes with identifiable rootlet fibers were detected in the anterior region of the KV. We now describe this observation descriptively and explicitly discuss the associated caveats, as noted above. Importantly, when indicating the locations of identifiable centrosomal structures across the dataset, we do not infer functional significance; rather, we report solely where such structures could be confidently resolved within

the vEM volume.

“similar concern here. Finding 2 cilia with CADVs doesn't mean they weren't missed in other regions so it's premature to say these are regionalized. Similarly, it does appear that more CaVs were detected in posterior parts of KV, but there isn't a clear L/R bias in the anterior. While this would fit current models about LRO function, I don't think the data here is strong enough to support this conclusion.”

This analysis represents a single static time point from one reconstructed KV. Across all segmented cilia, we identified only two instances of cilia-associated dense vesicles. We report these observations descriptively, without drawing conclusions regarding their function or spatial significance, and simply indicate where these structures were detected within the overall KV architecture. This example illustrates the type of ultrastructural information that can be obtained using this approach, while acknowledging the practical limitation that large sample sizes are difficult to achieve with vEM. The dataset also highlights the presence of at least two morphologically distinct vesicle types capable of associating with KV cilia.

Reviewer 2

“Even though interesting, the study remains superficial and is unclear in its focus and aims. “

We agree. The manuscript has been fundamentally restructured to focus on vEM methodology, segmentation strategy, and ultrastructural annotation, rather than mechanistic or developmental conclusions.

1. About the optogenetic method

** The CRY2/CIB1 IFT88 optogenetic system produces a statistically significant drop in the proportion of ciliated KV cells ($\approx 50\%$ reduction) and in cilium length, together with a marked decrease in lumen area ($p < 0.0001$). Only two experimental conditions (global vs KV specific clustering) are shown; additional controls such as a non binding CRY2 mutant or CIB1 without IFT88 would strengthen the claim of specificity and CRY2/CIB1 IFT88 without 488nm illumination.*

** The optogenetic system primarily traps the CIB1-RFP-IFT88 fusion protein, but because IFT88 functions as part of a multi-protein IFT-B complex, clustering the fusion protein also pulls endogenous IFT88 (and other IFT-B components) out of circulation. What is the proportion of endogenous IFT88 that is trapped. How does this treatment affect the non-ciliary function of IFT88? Can the authors show IFT88 stainings in these conditions?*

** Does the optogenetic treatment lead to left right defects? This should be documented.*

** What is the role of IFT88 in lumen formation? There is extensive literature addressing the mechanism of KV morphogenesis, including lumen formation.”*

We have removed the optogenetics data sets from the manuscript with the hopes to expand this part of the story based on reviewers comments and just focus on the vEM dataset as a useful methodology to better characterize KV cell morphology.

2. About the vEM analysis

“ The vEM analysis is based on a single fully segmented KV (67 cilia), which limits the generality of the reported centriole heterogeneity (70.1 % with both centrioles, 12 % lacking centrioles). Replicating the segmentation on multiple embryos and different stages is necessary. It would increase confidence and would constitute a great resource for the community.”

We agree and explicitly state that the dataset represents a single, fully segmented KV (top of page 11). Accordingly, the manuscript is positioned as a proof-of-principle resource that emphasizes depth of ultrastructural analysis rather than generality. Although our original goal was to extend this approach across multiple developmental stages and specimens, the technical challenges associated with reliably identifying and orienting the KV for sectioning, particularly following fixation, staining, and X-ray microscopy, render such analyses highly labor intensive (see Figure 2). While these practical constraints limited further expansion within the scope of this study, we believe that making this dataset and analytical framework openly available will

provide a valuable resource for the community. We therefore present this work as a methods-focused resource paper, with the intention that others may build upon and extend these analyses in future studies.

“The presence of three cilia sub-populations—those with both centrioles, only a mother centriole, or no detectable centriole (~12% of cilia)—is interpreted as evidence for progressive centriole elimination during KV maturation. However, this conclusion is based on a single time point (12 hpf) and specimen, making it difficult to determine whether the observed heterogeneity reflects a developmental sequence or intrinsic cell-to-cell variability. To solidify the elimination model, additional experiments such as immunostaining for PCM markers and temporal analysis across different developmental stages are needed to track centriole and appendage dynamics.”

This interpretation has been removed. The data are now presented as static ultrastructural heterogeneity observed at a single developmental time point, without invoking developmental progression. While we note this as a potential avenue for future investigation, we do not draw conclusions regarding centriole duplication, maturation, or elimination in the current study. In this context, we note that ongoing efforts in the laboratory aim to identify centriole-specific markers compatible with zebrafish to enable analysis of centriole dynamics. At present, the pericentriolar material (PCM) markers that reliably label centrosomes by immunofluorescence in our system include γ -tubulin. Notably, this marker is consistently detected at the base of cilia. γ -tubulin immunostaining is shown in Figure 1 and supports the interpretation that the centrosome-associated dense regions (CaDRs) observed in the vEM dataset correspond to PCM-associated structures present at ciliated bases.

“The functional relevance of the two vesicle subtypes (CaVs enriched posterior left, CaDVs confined to the right) is interesting, yet no statistics is presented. Is there a specific accumulation of these CaVs in asymmetric localisation of the KV. This needs to be addressed to enlarge the scope of the study.”

Functional interpretations have been removed. Vesicle subtypes are now described solely as morphologically distinct structures observed by vEM, with their positions noted descriptively. We note that the identification of cilia-associated vesicles in this context highlights opportunities for future investigation, and, to our knowledge, represents the first report of vesicles directly associated with KV cilia.

“The overall analysis should take into account the left-right function of the KV and carefully analyse evidences of asymmetries in its structure.”

This consideration motivated our initial presentation of the spatial positions of cilia associated with ciliary vesicles (Figure 5H). However, as both reviewers correctly noted, without additional specimens or complementary approaches, such as immunostaining or live-cell imaging, to increase sample size, it is not appropriate to draw conclusions regarding left-right function. At present, we have not identified antibodies or genetic markers that reliably label these structures for such analyses. We therefore present these observations as a descriptive starting point for future hypothesis generation and emphasize that this analysis derives from a single embryo. We argue that making this dataset and analytical framework available to the community, with these caveats stated, will facilitate further investigation and extension of these findings.

“Cite earlier work on distal and sub distal appendages when discussing the new appendage statistics.”

Done.

“Deposit the vEM dataset and segmentation scripts in an open repository to enhance reproducibility.”

All datasets will be deposited upon publication.

“Provide full statistical details (sample size per clutch, exact tests) in Table S1.”

Done.

“Giving the stage how the embryo in hours is not always robust as a stage indication (it fluctuates with temperature and sometimes with the ZF background). How many somites do the embryo display at this stage in the lab condition.”

We carefully screened 10-15 embryos for vEM processing, selecting specimens at the 6-somite stage (~12 hpf) with KVs containing between 50 and 100 cells, a range previously reported to support robust left-right asymmetry.

First decision letter

MS ID#: bio.062489

MS Title: Insights into the zebrafish Left-Right Organizer's Centrosomes and Cilia via Volume Electron Microscopy

Authors: Favour Ononiwu, Albert Lawrence Adhya, Melissa Mikolaj, Christopher Dell, Abdalla Wael Shamil, Kedar Narayan and Heidi Hehnly

I am happy to tell you that your manuscript has been accepted for publication in Biology Open, pending our standard publication integrity checks. It was accepted on 4th February 2026.